# It Takes a Village of Chromatin Remodelers to Regulate rDNA Expression

**DOI:** 10.3390/ijms26041772

**Published:** 2025-02-19

**Authors:** Mathieu G. Levesque, David J. Picketts

**Affiliations:** 1Regenerative Medicine Program, Ottawa Hospital Research Institute, Ottawa, ON K1H 8L6, Canada; mleve106@uottawa.ca; 2Department of Biochemistry, Microbiology, and Immunology, Faculty of Medicine, University of Ottawa, Ottawa, ON K1H 8M5, Canada; 3Department of Cellular and Molecular Medicine, Faculty of Medicine, University of Ottawa, Ottawa, ON K1H 8M5, Canada; 4Department of Medicine, Faculty of Medicine, University of Ottawa, Ottawa, ON K1H 8M5, Canada

**Keywords:** ribosome biogenesis, chromatin remodeling, ribosomopathy, neurodevelopmental disorder, nucleolus

## Abstract

Ribosome biogenesis is one of the most fundamental and energetically demanding cellular processes. In humans, the ribosomal DNA (rDNA) repeats span a large region of DNA and comprise 200 to 600 copies of a ~43 kb unit spread over five different chromosomes. Control over ribosome biogenesis is closely tied to the regulation of the chromatin environment of this large genomic region. The proportion of rDNA loci which are active or silent is altered depending on the proliferative or metabolic state of the cell. Repeat silencing is driven by epigenetic changes culminating in a repressive heterochromatin environment. One group of proteins facilitating these epigenetic changes in response to growth or metabolic demands are ATP-dependent chromatin remodeling protein complexes that use ATP hydrolysis to reposition nucleosomes. Indeed, some chromatin remodelers are known to have indispensable roles in regulating the chromatin environment of rDNA. In this review, we highlight these proteins and their complexes and describe their mechanistic roles at rDNA. We also introduce the developmental disorders arising from the dysfunction of these proteins and discuss how the consequent dysregulation of rDNA loci may be reflected in the phenotypes observed.

## 1. Introduction

Ribosomes are a critical component for cellular function as they are the molecular machinery required for the synthesis of proteins. Ribosome biogenesis begins with the transcription of the rDNA in the nucleolus followed by an energy-intensive and nucleolytic practice that requires ~200 accessory proteins to rapidly process the nascent transcript and then assemble the mature small and large subunits of the ribosome [1]. Moreover, different levels of ribosome production must be coordinated to facilitate other processes, such as cell proliferation, cell fate decisions, or general metabolic activity, that meet the needs of individual cell types [2]. Given this critical role, it is not surprising that aberrant ribosome biogenesis has been associated with diseases, commonly referred to as ribosomopathies, which include cancer susceptibility and growth, aging-related diseases, and neurodevelopmental disorders [1]. Many of these disorders affect the initial steps of ribosome biogenesis, namely rDNA transcription, thereby highlighting the importance of regulating the transcriptional output of this locus [2,3]. There are many mechanisms which regulate this process, including genomic elements at rDNA arrays, cell signaling pathways, various transcription factors, and epigenetic processes that regulate the fraction of active versus inactive rDNA repeats. In this review, we focus on the contributions of different chromatin remodeling complexes and their associated co-factors in mediating these early steps of ribosome biogenesis. We also highlight the alterations that occur when these proteins are dysfunctional, demonstrating the importance of proper regulation and maintenance of the chromatin architecture and drawing a link between rDNA transcriptional defects and the neurodevelopmental disorders arising from defects in these genes as a novel subset of ribosomopathies (Table 1).

## 2. Structural Organization and Transcription of the rDNA Repeats

Ribosomes are encoded in part by the 47S rDNA arrays, which in humans are located on the p-arms of the five acrocentric chromosomes, namely chromosomes 13, 14, 15, 21 and 22 [4]. Between the five chromosomes there are a total of several hundred copies (300–500 in humans) of the 47S array, each oriented in the same direction; this is an organization that is conserved in other species (e.g., mice) (Figure 1). Despite the large number of copies, many are in a transcriptionally silent state with a variable number of active genes [5]. Indeed, the repetitive nature of the rDNA arrays make them prone to recombination, and they require constant regulation to maintain their chromatin structure and maintain genomic integrity.

Most of the transcription that occurs in a mammalian cell is the transcription of rRNA. It has been shown that across eukaryotic organisms, rRNA makes up as much as 80–90% of the total RNA present in the cell [6]. Each 47S array encodes the 18S, 5.8S, and 28S rRNAs, and they are transcribed by RNA polymerase I. The rDNA arrays in mammals are structured as ~13 kb coding regions separated by intergenic spacer (IGS) regions which are ~30 kb in length (Figure 1). The smallest rRNA transcript, the 5S rRNA, is encoded by one locus on chromosome 1 in humans and is transcribed uniquely by RNA polymerase III [7]. Following transcription, these rRNAs are processed and assembled with ribosomal proteins to form functional ribosomes.

### 2.1. Genomic Elements and Cell Signaling Pathways

At the level of the genome, rDNA contains several regulatory elements which are necessary for the maintenance of the repeat. The mammalian IGS bears a replication fork barrier (RFB) site at the end of the 47S coding sequence, to which the transcription termination factor 1 protein (TTF-1) binds and terminates both transcription and replication (Figure 2) [8,9]. A promoter in the IGS region has been identified in mice, from which a long non-coding RNA is transcribed, which itself functions in regulating the transcription of the 47S transcript [10]. A recombination enhancer element, E-pro, exists in budding yeast and is a bi-directional promoter of a noncoding RNA which helps to maintain the high copy number of rDNA [11]. E-pro promotes gene amplification after deletional recombination via the dissociation of cohesin from DNA, which is normally present for equal sister chromatid recombination [12]. Such an element to enhance recombination has not been identified in mammals, although some mechanism likely exists to maintain a high copy number. DNA methylation is a major form of the epigenetic regulation of the rDNA loci and is critical for the repression of these genes. A significant portion of rDNA genes are silenced at any given time in lineage-committed cells; for instance, about half of the rDNA is methylated in NIH3T3 cells [10]. Methylation of a cytosine at position -133 relative to the 47S transcription start site (TSS) is enough to inhibit upstream binding factor (UBF) binding and therefore formation of the transcription initiation complex [13]. Beyond genomic elements and the epigenome, ribosome biogenesis is also subject to regulation via various cell signaling pathways. A well-known example of this is the mTOR signaling pathway, where the mTOR complex 1 can activate the S6K1 kinase, which in turn phosphorylates and activates UBF as well as the RNA polymerase I transcription initiation factor RRN3 (Figure 2) [14,15]. Transduction of growth stimuli via the MAPK signaling pathway also results in phosphorylation of RRN3 (Figure 2) [16].

### 2.2. Transcription Factors Involved in rDNA Transcription

Since transcription of rDNA arrays is required to make functional ribosomes, it is crucial that the chromatin architecture at the rDNA loci remains dynamic and carefully regulated to facilitate differential rDNA expression depending on cellular needs. Various transcription factors (TFs) are now known to be required at rDNA, especially for initiating and maintaining a high level of transcription. UBF was one of the first major transcription factors to be identified as having a role in the activation of rRNA transcription [17,18,19]. UBF interacts with selective factor 1 (SL1), which is part of a TF complex consisting of TATA-binding protein (TBP) and associated factors which confer promoter recognition specifically to RNA polymerase I to stimulate transcription (Figure 2) [20,21]. Intriguingly, UBF gain-of-function mutations in humans result in aberrant hyperactive rRNA production and childhood neuronal degeneration [22]. TTF-1 is another crucial rDNA transcription factor which was first identified as a factor mediating the termination of RNA polymerase I-dependent transcription [23,24]. Later studies demonstrated the ability of TTF-1 to activate rRNA transcriptional activation in cell-free transcriptional assays [25]. More recently, TTF-1 has been found to have a role in establishing a repressed chromatin environment at rDNA via recruitment of the nucleolar remodeling complex (NoRC), the main complex involved in forming heterochromatin at rDNA [26]. Probably the most intensely studied rDNA transcription factor is the oncogene c-Myc. Not only does c-Myc coordinate transcription of RNA polymerase I- and II-dependent genes, but also the transcription of RNA polymerase I-dependent genes [27,28,29]. c-Myc is a regulator of higher-order chromatin structure at rDNA, where it can induce the formation of chromatin loop structures and promote the recruitment of TTF-1 [30]. The ability of c-Myc to upregulate ribosome biogenesis as a rDNA TF likely plays a role in the protein driving oncogenesis [31,32].

### 2.3. Introduction to Chromatin Remodelers at the rDNA Locus

Chromatin remodelers are a class of proteins which use ATP hydrolysis to facilitate histone variant exchange or nucleosome repositioning, and many are known to be important in regulating the chromatin environment at rDNA loci (Table 1). There are four main classes of ATP-dependent chromatin remodelers which have been identified: Switch/sucrose non-fermentable (SWI/SNF), imitation switch (ISWI), chromodomain helicase DNA-binding (CHD), and inositol-requiring 80 (INO80). Perhaps the best-studied chromatin remodeling complex is the human BAF (BRG1/BRM-associated factors) complex, which is the highly conserved ortholog to the yeast SWI/SNF chromatin remodeling complex. In humans, the BAF complex is composed of two core interchangeable ATPase subunits, BRG1 (*SMARCA4*) or BRM (*SMARCA2*) and many other subunits, totalling roughly 15 proteins [33,34]. It is surprising to note that the BAF complex has not been shown to have a significant role in regulation of the rDNA locus. ISWI chromatin remodelers in humans are similarly conserved from yeast composed of one of two mutually exclusive ATPase subunits, SNF2L (*SMARCA1*) or SNF2H (*SMARCA5*), which can each pair with one of seven different regulatory subunits [35]. Thus, 14 different human ISWI complexes potentially exist, but only six have been characterized thus far: ACF, WICH, NoRC, NURF, CERF, and RSF [35,36]. There are nine known CHD ATPase proteins in humans, which are divided into three subfamilies based on their protein structure: I (CHD1/2), II (CHD3/4/5), and III (CHD6/7/8/9) [37]. These again function in multimeric protein complexes, the most well-characterized being the nucleosome remodeling and deacetylase (NuRD) complex [37]. Finally, the INO80 family of chromatin remodelers consists of two ATPases, namely INO80 and SRCAP in humans [38]. These proteins also both function in multimeric complexes which have been implicated in numerous chromatin regulation functions, but no role has yet been identified at rDNA for this class of chromatin remodelers [38]. “Orphan” chromatin remodelers which do not clearly belong to any of the four major families have also been identified (e.g., ATRX) [33,34]. An array of diseases and disorders have been linked to mutations in genes encoding the different classes of chromatin remodelers [33,34,35,36,37,38,39,40,41,42,43,44,45]. Pathologies which occur because of mutations in chromatin remodelers vary widely in phenotype, highlighting the diversity of biological processes that these proteins regulate. Chromatin remodelers are commonly mutated in cancers as well as diseases which disrupt growth and development [43,44,45]. As many as 20% of all cancers carry mutations in the protein subunits which make up the SWI/SNF complex [43]. Of all the genes identified with a link to intellectual disability (ID) and autism spectrum disorder (ASD), 10% encode chromatin modifying, binding, or remodeling proteins [46]. Often, these pathologies will present with significant defects in cellular proliferation and/or differentiation.

Perhaps the best-studied chromatin remodeling protein involved in rDNA regulation is Cockayne syndrome B (encoded by the gene *ERCC6*), which is the major gene that is mutated in Cockayne syndrome (CS) [47]. Multiple studies have shown reduced pre-rRNA synthesis upon loss of function of the protein [48,49]. Loss of function of the SWI/SNF-like chromatin remodeler ATRX (the causative gene in the ATR-X syndrome) results in enhanced replication stress and a significant impairment in neural progenitor growth, causing a reduction in neuron production and the size of the cerebral cortex [50,51]. Deletion of the ISWI gene *SMARCA5* (which encodes the SNF2H protein), a component of the NoRC complex, results in the inability of hematopoietic stem cells to differentiate into mature erythroid and myeloid cells, as well as severe proliferation defects within neural progenitors [52,53]. The nucleosome remodeling and deacetylase (NuRD) complex, harboring several members of the CHD family of chromatin remodelers, participates in neural progenitor expansion and differentiation [39]. Intriguingly, each of these critical chromatin remodelers (and others) have been shown to be recruited to rDNA and to modulate the transcriptional activity of this locus (Table 1) [54,55,56]. In the following sections, we discuss in more detail the role of individual chromatin remodeling complexes in the regulation of rDNA transcription and chromatin structure and introduce the neurodevelopmental disorders caused by dysfunction of these same complexes.

**Table 1 ijms-26-01772-t001:** Summary of chromatin remodeling proteins and complexes involved in rDNA regulation and diseases associated with mutations in these complexes.

Chromatin Remodeling Protein/Complex	Action at rDNA	Disease(s) Associated	Clinical Manifestations
NuRD	Maintains a poised, transcriptionally permissive state [56]	Snijders Blok–Campeau syndrome (SNIBCPS) (*CHD3*) and Sifrim–Heitz–Weiss syndrome (SIHIWES) (*CHD4*) *CHD5* mutations result in a novel neurodevelopmental disorder [57,58,59]	Intellectual disability, developmental delay, facial dysmorphisms [57,58,59]
NoRC	Promotes heterochromatin formation to silence transcription; propagates H3K9me3 and H3K27me3 [60]	*SMARCA5* mutations result in a novel neurodevelopmental disorder [61]	Developmental delay, microcephaly, short stature [61]
CSB	Mediates fully active transcription, positively regulates transcription [62]; maintains genomic stability by resolving G4 quadruplexes [63]	Cockayne syndrome [64]	Microcephaly, intellectual disability, dwarfism [64]
ATRX	Maintains heterochromatin stability possibly through deposition of H3.3 or binding G4 quadruplexes and R-loops [42,65]	ATR-X syndrome [45]	Intellectual disability, hypotonia, facial dysmorphisms [66]

## 3. Defined Roles for Different Chromatin Remodeling Complexes in rDNA Regulation

### 3.1. Facilitation of rDNA Transcription by CSB

Initially characterized as an ATP-dependent chromatin remodeler involved in transcription-coupled nucleotide excision DNA repair (TC-NER), CSB has now emerged as an important factor in ensuring efficient rRNA transcription [49,62]. CSB is a SWI/SNF-like chromatin remodeler, possessing the seven highly conserved ATPase motifs characteristic of these proteins [67].

Broadly, CSB is known to be a regulator of transcription by RNA polymerase I, II, and III [68,69]. The effect of loss of function of CSB on global transcription is dramatic, with CSB-deficient cells having as much as a 50% lower level of RNA synthesis versus normal cells [70]. Loss of CSB was shown to result in metaphase fragility of the U1, U5, and 5S genes, possibly because of RNA polymerases remaining on DNA at metaphase, thus impeding the local condensation of chromatin. This metaphase fragility was rescued upon CSB re-expression [69].

CSB is a chromatin remodeler which has a distinct role in the activation of rRNA transcription rather than suppression. Depletion of CSB reduces RNA polymerase I association with rDNA, while overexpression of CSB enhances the production of rRNA in an ATPase-dependent manner [49]. CSB can interact directly with TTF-1 to be recruited to the promoter-proximal sequence of rDNA (Figure 3A) [49]. CSB was further found to interact with the histone methyltransferase G9a, which is responsible for the mono- and di-methylation of H3K9 in euchromatic regions. Surprisingly, although H3K9 methylation is usually associated with transcriptional repression, CSB recruited G9a to active rDNA regions, and the methylation activity of G9a was required for efficient RNA polymerase I transcription (Figure 3A) [49]. A more recent study has established that CSB and another TC-NER protein, Cockayne syndrome A (CSA, encoded by the ERCC8 gene), both interact with the abundant nucleolar protein nucleolin [60]. In the case of CSB, its ubiquitin binding domain is required for this interaction. Strikingly, this study further showed that the transcriptional activation of rDNA by CSB appeared to be dependent on nucleolin, as CSB WT cells depleted of nucleolin showed a significant decrease in 47S rRNA output. This study suggests a cooperation between all three factors in ensuring efficient pre-rRNA synthesis and RNA polymerase I transcription elongation [60]. Finally, CSB can also facilitate rRNA transcription via the resolution of DNA secondary structures that can obstruct transcription (Figure 3B). Due to its repetitive nature and high GC content, rDNA is particularly prone to forming secondary structures, especially G4 quadruplexes [71]. G4 quadruplexes arise from the folding of G-rich sequences into hydrogen-bonded planar quartets which can then stack on top of one another [72]. CSB can effectively melt these structures on its own in an ATP-independent manner. In CSB deficient cells, RNA polymerase I had increased stalling at putative G4-forming sequences, suggesting an increased difficulty in transcribing these sequences in the absence of CSB [63].

As its name suggests, mutations in the CSB protein are the direct cause of CS in humans. CSA mutations also manifest as CS but occur much less frequently than CSB mutations [47]. CS is an ultra-rare disease, occurring in approximately 1 out of every 250,000 live births [64]. Common clinical presentations include photosensitivity, intellectual disability, hearing loss, dwarfism, and genital abnormalities, among others [64]. These hallmarks of CS are striking in that there is overlap with many other disorders where RNA polymerase I transcription is significantly affected [47]. CS currently has no treatment available. Only symptomatic care is an option, consisting of physical therapy, cochlear implants, and feeding tubes, among others. The median age of death for those afflicted is 12 years of age [64]. CS is an example of a growing group of accelerated aging diseases. Mutations in CSB are spread across the protein with all types of mutations being represented in the cases identified so far. There is a group of mutations clustered in the ATPase motif, highlighting the importance of the chromatin remodeling function of CSB [47]. While it was originally thought that defective TC-NER from CSB mutation could explain the phenotype of the disease, it is becoming apparent that this cannot explain all aspects of CSB. Other diseases of defective TC-NER have significant differences in clinical presentation from CSB, such as xeroderma pigmentosum, where there is a ten thousand-fold increase in the risk of skin cancers, which is not found in those with CSB [73]. Taken together, it is likely that dysfunction of RNA polymerase I-mediated transcription is a yet unappreciated determinant of the phenotype of CSB and other similar diseases.

### 3.2. Priming of the rDNA Locus by the NuRD Complex

NuRD is a major ATP-dependent chromatin remodeling complex in eukaryotic organisms. The initial role described for NuRD was as a transcriptional repressor. This was proposed because the complex is unique in its ability to not only slide nucleosomes along DNA but also to deacetylate histones, rendering chromatin less accessible [74]. It is now understood that the complex is multifaceted and localizes to the promoters of highly transcribed genes as well [75].

Compositionally, the NuRD complex is built from six different families of proteins (Figure 4). The core ATP-dependent chromatin remodeling component of NuRD is the CHD protein, specifically CHD3, CHD4, and CHD5, which function as the catalytic subunit [76,77]. The histone deacetylase activity of NuRD arises from the integration of histone deacetylase 1 (HDAC1) or HDAC2 to the complex [78]. Methyl-CpG-binding domain (MBD) proteins provide affinity to methylated DNA for the complex [79]. Retinoblastoma-binding protein (RBBP) 4 and 7 provide histone tail binding capability; some groups have reported an association of GATA2A and GATA2B with NuRD, which are also known to have a high affinity for unacetylated histone tails [79,80,81]. Finally, metastasis-associated genes (MTA) 1, MTA2, and MTA3 of the NuRD complex are known to interact with transcription factors which mediate transcriptional repression [82].

NuRD was recently shown to be an important regulator of developmental processes [83,84,85]. This is especially evident throughout mammalian corticogenesis, where subunit exchanges between CHD4, CHD5, and CHD3 are critical for neural progenitor expansion, neuronal migration, and differentiation, respectively [83]. In mice and zebrafish, NuRD is required for hematopoietic stem cell maintenance and differentiation [84,85].

There is mounting evidence that NuRD plays an important role in controlling the transcription of rRNA, which may be important in the manifestation of NuRD-related diseases. At the rDNA locus, TTF-1 localizes to a consensus motif 170 base pairs upstream of the transcription start site. Here, TTF-1 can recruit NuRD; this was determined by the finding that CHD4 interacts directly with TTF-1 [56]. Intriguingly, the CHD4-containing NuRD complex is important for maintaining the activation of rRNA synthesis as depletion of CHD4 results in a significant impairment of RNA polymerase I binding to rDNA, a reduction in rRNA synthesis, and an increased level of DNA methylation at the locus (Figure 4) [56]. The chromatin remodeling and histone binding activity of CHD4 is required for the transcriptional activation of rRNA. Deletion mutants of CHD4 in the chromodomain and ATPase domain overexpressed in CHD4-depleted cells were unable to rescue the transcriptional deficit incurred upon CHD4 loss [56]. An interesting possibility suggested by this study is that the NuRD complex operates in a coordinated fashion with CSB to activate transcription of rRNA. Tandem affinity purification of CSB protein co-purified all the protein subunits of NuRD [56]. Other studies on CSB have shown that it can recruit p300/CBP-associated factor (PCAF) to rDNA promoters. PCAF is a histone acetyltransferase, and histone acetylation at the rDNA promoter is required for the assembly of the RNA polymerase I transcription initiation complex [86]. As such, the authors proposed that the CHD4-containing NuRD complex establishes the poised state of the rRNA gene, and CSB can then mediate the transition from the permissive to the fully active state [56]. In addition, NuRD can indirectly maintain the demethylation state of the rDNA loci. NuRD binds directly to the promoter of TTF-1 interacting protein 5 (TIP5), the regulatory subunit of the ISWI-containing NoRC complex, and represses its transcription in mouse embryonic fibroblasts (MEFs) [87]. During reprogramming of MEFs into induced pluripotent stem cells (iPSCs), the expression of CHD4 decreased concurrently with increased expression of TIP5, thereby increasing the fraction of silenced rDNA genes [87]. Similarly, the MBD3 subunit of NuRD can epigenetically regulate the rDNA independently of NuRD. MBD3 was shown to bind directly to the same rDNA promoters as UBF [88]. Overexpression of MBD3 led to increased demethylation of the rDNA promoter, while depletion of MBD3 had the opposite effect [88]. Overall, NuRD is responsible for establishing a specific chromatin state at rDNA, which results in the gene being transcriptionally inactive but poised for transcriptional activation, thus remaining accessible for transcription factor binding [56].

All the CHD proteins that can participate in the NuRD complex are pathogenic in humans when mutated in the germline. CHD3 mutations are causative in Snijders Blok–Campeau syndrome (SNIBCPS), a rare developmental disorder with varying clinical presentation, except that all patients have some developmental delay or intellectual disability [57]. CHD4 mutations cause Sifrim–Heitz–Weiss syndrome (SIHIWES), which has similar clinical characteristics to SNIBPCS, with most patients having speech delay and mild to moderate intellectual disability [58]. Finally, CHD5 mutations have recently been identified as causing a neurodevelopmental disorder that is clinically very similar to those associated with CHD3 and CHD4 mutations [59]. Mutations in the pathogenic gene variants tend to cluster in the ATPase-helicase domain, thus affecting the chromatin remodeling ability of the proteins. Most of the other known mutations are located in the other functional domains of the proteins, especially the chromodomain, which is crucial for histone tail recognition [57,58,59]. Mouse models of loss of function of CHD4 and CHD5 proteins result in significant disruption of gene expression, especially of genes pertaining to cellular growth and proliferation [83,89]. No direct link to misregulated rRNA transcription has been made between the phenotype of these diseases in humans and their mouse model counterparts. However, the clinical presentation of these diseases has similarities to others where this link has been made, perhaps the most notable example being CS, as previously discussed.

### 3.3. Maintenance of rDNA Repressive Heterochromatin by NoRC

NoRC is an ISWI-containing chromatin remodeling complex, so named because of its apparent strong localization to the nucleolus upon its initial characterization and interaction with TTF-1 [90]. The protein subunits of NoRC include TTF-1 interacting protein 5 (TIP5, also known as BAZ2A) and SNF2H (Figure 5) [90]. NoRC is described as the primary chromatin remodeling complex responsible for silencing the rDNA loci. NoRC recruitment to the rDNA loci results in the accumulation of repressive heterochromatin marks, including trimethylation of lysine 9 on histone H3, as well as hypoacetylation of histone H4 [91]. Intriguingly, the interaction between TIP5 and TTF-1 is via the same TTF-1 domain which interacts with CSB, suggesting competition between the two proteins in determining the epigenetic state of rDNA. Indeed, experiments overexpressing either TIP5 or CSB showed that overexpression of one protein will result in less binding of the other protein to TTF-1 [49]. TIP5 is a large protein which possesses an MBD-like domain and a PHD adjacent to a bromodomain (BRD), also known as a bromodomain adjacent to zinc finger (BAZ) domain. The protein also contains a TIP5/ARBP/MBD (TAM) domain responsible for binding dsDNA and dsRNA [92]. The structural similarity between TIP5 and the related protein Williams syndrome transcription factor (WSTF, also known as BAZ1B) is notable, as WSTF forms a distinct chromatin remodeling complex with SNF2H, known as B-WICH, which is also a regulator of rRNA genes [93]. The PHD domain of TIP5 can interact with other proteins that are important for establishing heterochromatic features, including DNA-methyltransferases (DNMTs), HDAC1, and a methyltransferase responsible for trimethylation of H3K9, SETDB1 (Figure 5) [94]. To facilitate NoRC recruitment to rDNA chromatin, the BAZ domain can bind H4K16ac, as well as have affinity for H3K14ac [91,92]. TIP5 also possesses an MBD-like domain that can bind dsDNA and dsRNA, although with respect to dsDNA, this occurs in a non-specific, methylation-independent manner [95]. Beyond the NoRC complex, TIP5 associates with large swaths of active genomic regions outside of the rDNA in mouse embryonic stem cells (mESCs) and appears to restrict active chromatin compartments [96]. TIP5 is in fact largely excluded from rDNA in ESCs and is recruited upon differentiation to establish heterochromatin at these regions [97]. A possible explanation for this is that ESCs require more open chromatin at rDNA to ensure that rDNA transcription remains high while the cells are proliferating and remain uncommitted. Once the cells differentiate and no longer need to divide, TIP5 is recruited to throttle rDNA transcription as the metabolic needs of the cell change.

NoRC is unique among chromatin remodeling complexes due to its specificity to the rDNA repeats and proximal heterochromatin. Elimination of the NoRC complex via siRNA-mediated knockdown of TIP5 in mouse NIH3T3 cells results in significant increases in rRNA production, with enlarged and fewer nucleoli versus control cells [91]. Transmission electron microscopy (TEM) revealed that condensed chromatin in contact with the fibrillar centre (FC) of the nucleolus, where repressed rRNA genes are normally located, was almost entirely lost without NoRC [91]. Loss of NoRC resulted in instability of the rDNA repeats, with a significant loss of the number of rDNA copies versus control cells. Repetitive DNA in proximity to rDNA (major and minor satellite DNA) was also lost, indicative of instability at these regions as well, and a possible role for NoRC in mediating heterochromatin spreading to these regions [91]. Through the suppression of rDNA expression, NoRC has also been found to limit cellular proliferation in ESCs. Interestingly, in these cells the histone variant H2A.X was responsible for the recruitment of the complex to rDNA loci [98]. TIP5 was able to interact directly with H2A.X, and without H2A.X, rRNA levels and cellular proliferation of the ESCs increased drastically. Strikingly, a TIP5 knockdown in cells which were already depleted of H2A.X resulted in little further increase in rRNA output or cellular proliferation, demonstrating the importance of H2A.X in NoRC recruitment [98].

Intriguingly, while no TIP5 mutations have been linked to human disease, haploinsufficiency of the very closely related protein WALp4 (also known as BAZ2B) has recently been found to be causative in a novel developmental disorder presenting with clinical features of intellectual disability and autism spectrum disorder [99]. The biochemical function of BAZ2B remains poorly understood. The protein is very similar in structure to TIP5, containing both TAM and BAZ domains [100]. BAZ2B can form a chromatin remodeling complex with SNF2H and its homolog SNF2L in vitro; however, the biological significance of this finding is yet to be determined [1]. This opens the intriguing possibility that BAZ2A/B are interchangeable in the NoRC complex in an in vivo context, thus providing a mechanistic link between the destruction of the NoRC complex via haploinsufficiency of BAZ2B and its disease phenotype. Furthermore, mutations in SNF2H have recently been found to be directly linked to disease in humans [61]. Pathogenic variants of SNF2H result in a neurodevelopmental disorder presenting with clinical features such as developmental delay, intellectual disability, and microcephaly [61]. Intriguingly, microcephaly is also seen in mice deficient in SNF2H, a somewhat paradoxical phenotype as greater proliferation would be expected with loss of function of NoRC [53]. Of the 12 individuals identified in the study linking SNF2H dysfunction to human disease, 7 had mutations in the ATPase domain, likely disrupting ATP hydrolysis and therefore the ability of the protein to slide nucleosomes and alter chromatin structure. While the precise mechanistic link remains to be elucidated, there is dysfunction of chromatin remodeling via NoRC when its constituents are mutated, and the resultant instability of ribosome biogenesis is a likely contributor to the phenotype that manifests.

### 3.4. Maintenance of Genomic Stability of rDNA by ATRX

Alpha-thalassemia mental retardation syndrome X-linked (ATRX) is unique among chromatin remodeling proteins as it does not fit neatly into the four main classes discussed previously [45]. ATRX is a SWI/SNF-like chromatin remodeler, possessing the classic ATPase/helicase domain of this family [42]. ATRX has several well-characterized protein and chromatin binding domains. At the N-terminal region, ATRX possesses an ATRX-DMNT3-DNMT3L (ADD) domain, which is the histone recognition module of ATRX specific to the H3K9me3 modification [101]. ATRX can also bind HP1α, a heterochromatin protein strongly enriched in H3K9me3 regions, via an N-terminal domain [102]. One of the best-characterized protein–protein interactions of ATRX is DAXX, which, when bound to ATRX, forms a histone chaperone complex specific to the deposition of the histone variant H3.3 (Figure 6) [103,104]. At its C-terminal region, ATRX has a promyelocytic leukemia protein (PML) targeting domain as well as a MeCP2 interaction motif [105,106].

The role for ATRX in maintaining heterochromatin integrity extends to the rDNA repeats. Broadly, ATRX is known for binding repetitive GC-rich chromatin in heterochromatic regions, including in telomeres, pericentromeric chromatin, and rDNA, to prevent replication stress [107,108]. ATRX can also bind the G4 quadruplexes which tend to form at these regions; however, it has not been demonstrated that it can directly resolve these structures in vitro (Figure 6) [108]. Beyond constitutive heterochromatin, ATRX can also interact with euchromatic regions and modulate the transcription of active genes [109]. ATRX-mediated transcriptional programming is important for determining cell fate, especially that of neurons [50,65].

In an early study, ATRX was found to localize to the short arms of metaphase acrocentric chromosomes [110]. In blood cells from patients with ATR-X syndrome, the rDNA repeat loci were severely hypomethylated compared to those from normal individuals [110]. In mESC ATRX KO models, as much as 50% rDNA copy number loss was observed [54]. Heterochromatic histone marks such as H3K9me3 and H4K20me3 were also significantly reduced simultaneously with an increase in γH2Ax levels at the rDNA loci. The deposition of H3.3 was also negatively impacted, likely disrupting ATRX-mediated heterochromatin assembly. Strikingly, due to a loss of heterochromatic silencing, the silent rDNA pool in the KO cells was almost completely absent, but at the same time, the overall rRNA transcriptional output was considerably reduced [54]. Another recent study focusing on ATRX mutants in neuroblastoma tumors found differences in rRNA expression depending on the ATRX mutation that was present. Neuroblastoma lines which expressed ATRX with exons 2–10 deleted showed downregulation of rRNA, while those that had ATRX with exons 2–13 deleted or were ATRX-negative had increased rRNA levels [111]. Other evidence of ATRX acting indirectly on rDNA or ribosome biogenesis comes from recent proteomics data using the BIOID system. In that study, the authors found ATRX to be in proximity to several proteins implicated in ribosome biogenesis [112]. ATRX interacted with FAM207A, a protein involved in maturation of the 40S ribosome subunit. IMP3 was also identified in the BIOID screen; it is a protein required for early pre-rRNA processing and is part of the U3 small nucleolar ribonucleoprotein (U3snoRNP) complex [113]. Altogether, the literature suggests an important role for ATRX in regulating rDNA, but its exact biological function at this locus remains largely uncharacterized.

**Figure 6 ijms-26-01772-f006:**
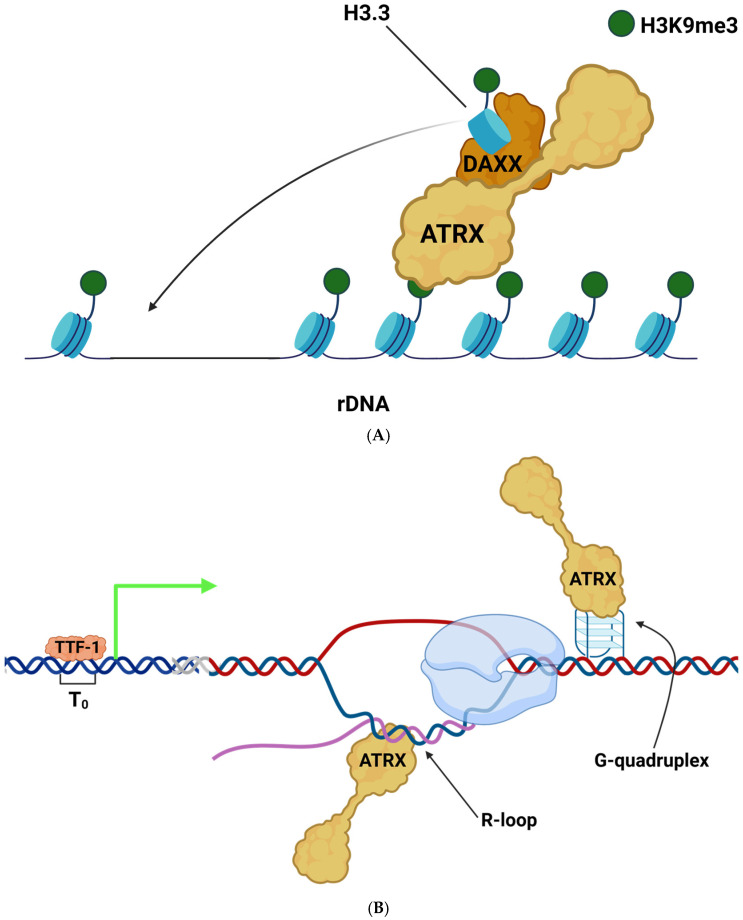
ATRX likely maintains chromatin integrity at rDNA. (**A**) ATRX forms a histone chaperone complex with DAXX, which deposits histone H3.3 containing the K9me3 modification at heterochromatic regions, including rDNA to maintain chromatin integrity [103,104]. (**B**) ATRX is known to bind and prevent the formation of G-quadruplexes and R-loops, which can form during DNA replication or transcription, resulting in genomic instability at the rDNA locus [108,114]. TTF-1 binding site is highlighted as the blue double helix in (**B**). Other DNA is represented as the white double helix or + (blue) and – (red) strands. RNA is represented as the purple strand in (**B**).

Pathological variants of ATRX in the germline cause ATR-X syndrome, a rare and severe neurodevelopmental disorder with an estimated rate of occurrence of about 1–9/1,000,000 [66,115]. The clinical manifestations of ATR-X syndrome are wide-ranging, with the only common manifestation between all patients being intellectual disability, which can range from mild to severe [116]. Other common clinical features include hematological abnormalities (including α-thalassemia), genital abnormalities, skeletal abnormalities, and gastrointestinal issues. While ATR-X is a debilitating disorder, some patients can live well into adulthood [66]. Only symptom-specific treatments are currently available. Pathological mutations in the germline tend to cluster within the ADD domain and the ATPase domain of ATRX, highlighting the functional importance of these domains in histone binding and chromatin remodeling, respectively. There have been 192 different germline mutations identified to date, the majority of which are missense or nonsense point mutations [116]. Beyond the germline, ATRX is also frequently mutated in some forms of cancer, including glioblastomas, pancreatic neuroendocrine tumors, and sarcomas [117,118,119]. Indeed, tumors with the alternative lengthening of telomeres (ALT) phenotype are highly correlated with ATRX loss of function, with approximately 90% of ALT tumors harboring ATRX mutation [120]. Loss of ATRX on its own is not enough to generate ALT de novo¸ suggesting there are other not yet understood factors at play [108]. Altogether, the loss of the histone binding and chromatin remodeling ability of ATRX is clearly crucial in driving a disease state, resulting from global changes in transcription and genome instability, especially at heterochromatic regions including rDNA.

## 4. Other Transcription Factors and Chromatin Regulators at rDNA

There are several other chromatin-modifying proteins and transcription factors which play a role in rDNA chromatin homeostasis in tandem with the chromatin remodelers and complexes that are less well characterized. The previously mentioned chromatin remodeling complex, B-WICH, is important for facilitating RNA polymerase III-mediated transcription [93]. B-WICH can form a large chromatin remodeling assembly with several nucleolar factors, including CSB and ribonucleoproteins [93]. Intriguingly, the chromatin remodeling activity of B-WICH is a requirement for the binding of c-Myc to the 5S rRNA locus [121]. While not chromatin remodelers per se¸ the RecQ helicases BLM and WRN are ATP-dependent DNA helicases which play a vital role in numerous DNA regulation processes, including maintaining genome stability and transcription [122,123,124,125]. Both proteins regulate RNA polymerase I-mediated transcription, but this function remains understudied. Both WRN and BLM stimulate rRNA transcription and are recruited to the nucleolus by rRNA transcription [126,127]. The rDNA repeats are particularly unstable in cells that have lost WRN or BLM protein function [128,129]. These proteins are implicated in Werner syndrome and Bloom syndrome, respectively, both very rare diseases occurring between 1 in 1,000,000 and 1 in 10,000,000 births [130,131]. While the two proteins are similar in structure and function, their related diseases are unique. Bloom syndrome is characterized by dwarfism, photosensitivity, and an extremely elevated risk of developing cancer [131]. On the other hand, Werner syndrome is considered a progeroid syndrome, with those afflicted presenting with similar manifestations of normal aging but in their 20s or 30s [130].

Aside from chromatin remodelers and the UBF and TTF-1 TFs, members of the plant homeodomain-like finger (PHF) family of proteins stand out as having a hand in transcriptional regulation of the rDNA loci. PHF6 and PHF8 both modulate rRNA transcription and are linked to two separate X-linked intellectual disability syndromes, Börjeson–Forssman–Lehmann syndrome (BFLS) and X-linked intellectual disability, Siderius type, respectively [132,133]. The two proteins are structurally closely related; however, PHF8 contains a Jumonji C-terminal (JmjC) domain which confers histone lysine demethylase activity, a role typically associated with transcriptional activation [134]. Conversely, PHF6 is known to bind repressive chromatin marks, such as H3K9me3 and H3K27me1, and recruit the histone methyltransferase SUV39H1 to the rDNA loci [135]. Consistent with this role, PHF6 was shown to suppress the transcription of rRNA, while PHF6 knockdown and PHF6 mutations resulted in elevated levels of rRNA production [135,136]. PHF8 knockdown models show a reduced expression of rRNA, while PHF8 overexpression results in upregulation of rRNA [137]. Like the other chromatin-modifying proteins discussed, both proteins have distinct roles in maintaining rRNA homeostasis, and pathogenic variants of each result in similar phenotypic outcomes in humans. Further work is required to dissect how exactly rRNA misregulation contributes to the clinical outcomes seen in these and other disorders discussed in this review.

## 5. Conclusions and Future Perspectives

In this review, we focused on the maintenance of rDNA transcription, but it should be noted that many chromatin remodelers also participate in DNA damage repair on rDNA; these have been addressed in several excellent reviews [131,138,139]. Maintaining the chromatin state of rDNA is at the heart of maintaining a homeostasis of ribosome biogenesis. Chromatin architecture at this locus is dynamic depending on cellular needs; thus, it necessitates a myriad of factors to maintain rRNA transcriptional homeostasis to preserve the ensuing downstream ribosome biogenesis. Major chromatin remodeling proteins and complexes are implicated in both the activation and repression of rDNA expression. NoRC is especially important for acting as a repressive force on the chromatin at rDNA, with the chromatin accumulating H3K9me3 in its wake and throttling rRNA output, ultimately resulting in reduced cellular proliferation (Figure 5). Others act as activators, such as NuRD and CSB, working together to establish a poised state and then the fully active state of the rDNA loci, respectively (Figure 3 and Figure 4). While not acting as a transcriptional activator per se, the chromatin remodeler ATRX maintains the integrity of the genome at rDNA repeats, ensuring that the copy number is not lost during replication and that rRNA output can thus be maintained (Figure 6). Still other proteins, including ATP-dependent helicases and non-enzymatic transcription factors, act as both repressors and activators to modulate rRNA transcription.

A common theme with all the proteins and protein complex subunits discussed in this review is that they are implicated in congenital human disorders, usually of a severe and incurable nature. Some key questions remaining are as follows: (1) How do the changes in rDNA expression caused by mutant remodeling complexes contribute to the phenotypes observed for these complex neurodevelopmental disorders? (2) Should altered rDNA chromatin be considered a novel class of ribosomopathies? (3) Can rRNA dysregulation represent a potential target for clinical management of these diseases? Strengthening the research on the precise links between chromatin remodeling at rDNA loci and its effect on ribosome biogenesis and disease pathogenesis is bound to help answer these questions and will prove fruitful in the exploration of avenues for the clinical management of these currently incurable syndromes. 

## Figures and Tables

**Figure 1 ijms-26-01772-f001:**
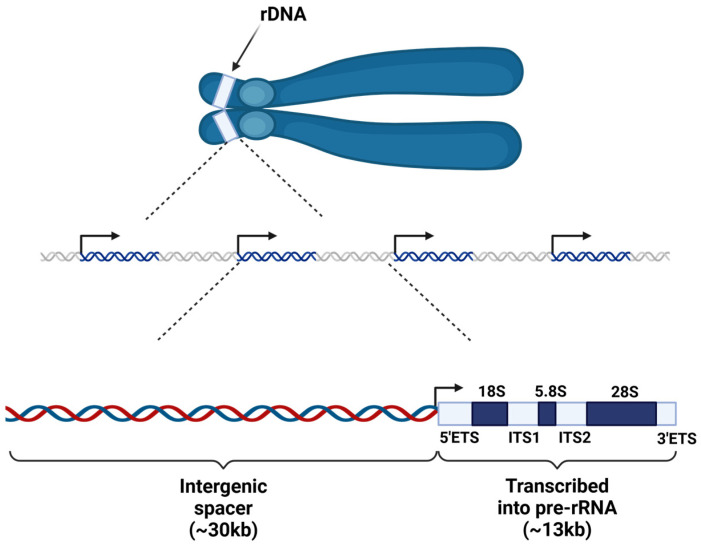
The genomic arrangement of the human rDNA genes. rDNA is encoded on the p-arms of the five acrocentric chromosomes (top cartoon) [4]. Here, many copies of the rDNA locus are arranged in arrays in a head-to-tail manner (middle cartoon). Each copy of the rDNA locus can be divided into a ~30 kb intergenic spacer region and a ~13 kb coding region (bottom cartoon). Included in the coding region are 5′ and 3′ externally transcribed spacers (ETS) and two internally transcribed spacers (ITS). The black arrows indicate the transcriptional start sites (TSS) of each rDNA repeat. DNA is represented as the blue and white double helices in the middle cartoon, with blue and white coloring indicating rDNA and spacer regions, respectively. In the bottom cartoon, DNA is represented as the + (blue) and – (red) strands.

**Figure 2 ijms-26-01772-f002:**
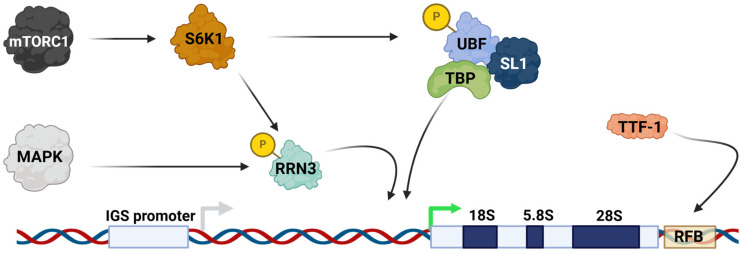
Transcription factor and cell signaling control of rRNA transcription. TTF-1 binds a replication fork barrier (RFB) near the 3′ end of the 47S coding sequence to aid in the termination of transcription and replication [8,9]. A promoter (IGS promoter) has been identified 2kb upstream of the pre-rRNA TSS (green arrow) in the mouse intergenic spacer sequence (IGS), which transcribes a long non-coding RNA that regulates pre-rRNA transcription [10]. In the transcriptional activation of rRNA, UBF forms a complex with SL-1, TBP, and its associated proteins to form a complex to stimulate RNA polymerase I-mediated transcription [13]. mTORC1 signaling can activate the S6K1 kinase, which in turn phosphorylates and activates UBF, as well as the RNA polymerase I transcriptional initiation factor RRN3 [14,15]. MAPK is also capable of phosphorylating RRN3 [16]. DNA is represented as the + (blue) and – (red) strands.

**Figure 3 ijms-26-01772-f003:**
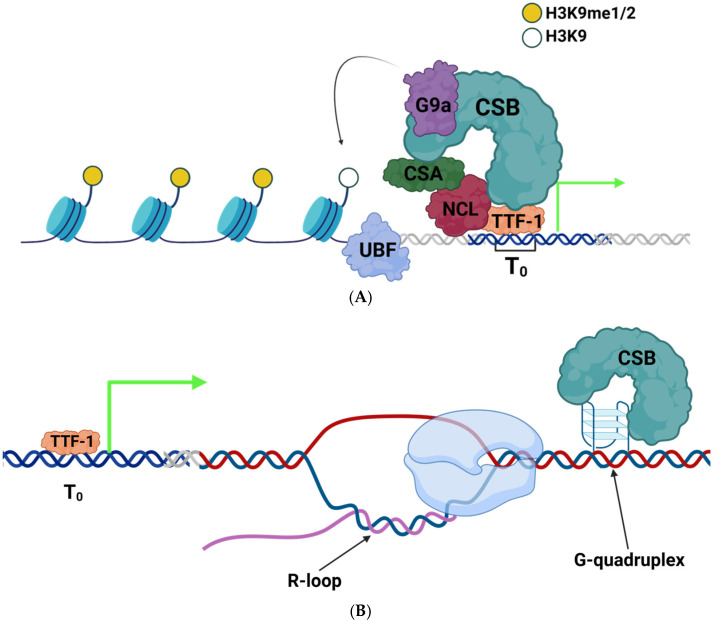
CSB acts as an activating factor in rRNA transcription. (**A**) CSB in complex with CSA and nucleolin (NCL) interacts with TTF-1 to mediate the chromatin state transition from poised to fully active for transcription by the addition of methyl groups to H3K9 via the interaction with G9a enzyme [49,56,60]. (**B**) CSB can also bind and resolve DNA secondary structures such as G-quadruplexes. G-quadruplexes impede polymerase progression (blue complex), leading to R-loops and genomic instability that impede rDNA transcript production [63]. Green arrows depict active transcription. TTF-1 binding site is highlighted as the blue section of the double helix in (**A**,**B**). Other DNA is represented as the white double helix or + (blue) and – (red) strands. RNA is represented as the purple strand in (**B**).

**Figure 4 ijms-26-01772-f004:**
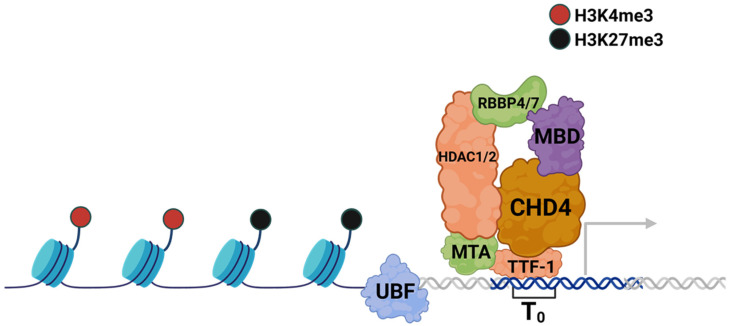
NuRD aids in activating rRNA transcription. The CHD4-bearing NuRD complex (CHD4, HDAC1/2, MBD, RBBP4/7 and MTA) is recruited by TTF-1 and maintains rDNA genes in a poised state, as indicated by the H3K4me3 and H3K27me3 bivalent marks [56]. TTF-1 binding site is highlighted as the blue double helix, with other DNA shown as the white double helix.

**Figure 5 ijms-26-01772-f005:**
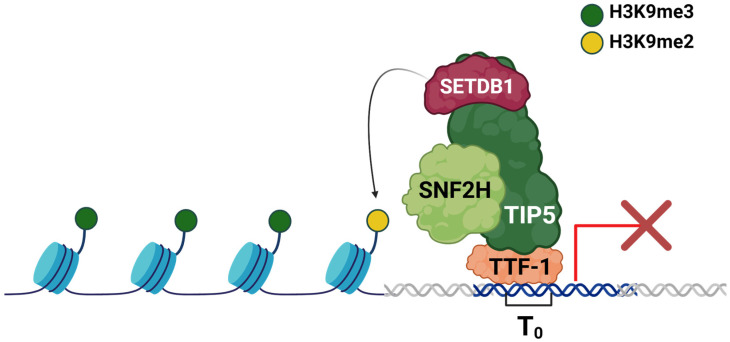
NoRC silences rRNA expression. The NoRC complex, comprising SNF2H and TIP5, can recruit the H3K9 methyltransferase SETDB1 to rDNA, which facilitates the methylation of H3K9me2 to H3K9me3, a mark of silenced heterochromatin [94]. Red X indicates that transcription is silenced. TTF-1 binding site is highlighted as the blue double helix, with other DNA shown as the white double helix.

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
