# Peer review of "It Takes a Village of Chromatin Remodelers to Regulate rDNA Expression"

_ijms, 2025, doi:10.3390/ijms26041772_

Round 1

Reviewer 1 Report

Comments and Suggestions for Authors

It takes a village of chromatin remodellers to regulate rDNA expression

by

Mathieu G. Levesque and David J. Picketts

This manuscript reviews the role of several ATP-dependent chromatin remodelling factors and related proteins in the regulation of the rDNA loci and rRNA expression. This is a well written and timely review from a lab that has been studying such factors for more than 10 years. I found this review interesting and informative. An important take-home message is the connection between neurodevelopmental disorders and miss-regulation of the rDNA loci because of mutations in genes coding for ATP-dependent chromatin remodelling factors.

I have only a few very minor comments that the authors might wish to address. I do not see a necessity to re-review.

Line 174-175: “Surprisingly, the SWI/SNF or BAF complex has not been shown to have a significant role in regulation of the rDNA locus”; briefly explain the connection between BAF and SWI/SNF here or in the preceding paragraph. Some readers may not know that BAF is a SWI/SNF type complex. BAF should also be listed in the abbreviation glossary.

In the cartoon figures (e.g., Figure 4), some of the protein names in the protein representations are hard to read, consider larger and bold font and maybe black instead of white. The histones (H3) in Figure 4 also look odd, like proteins sitting on top of DNA. A more ‘correct’ and appealing representation of nucleosomes with their histones might be good.

Author Response

Reviewer 1

This manuscript reviews the role of several ATP-dependent chromatin remodelling factors and related proteins in the regulation of the rDNA loci and rRNA expression. This is a well written and timely review from a lab that has been studying such factors for more than 10 years. I found this review interesting and informative. An important take-home message is the connection between neurodevelopmental disorders and miss-regulation of the rDNA loci because of mutations in genes coding for ATP-dependent chromatin remodelling factors.

I have only a few very minor comments that the authors might wish to address. I do not see a necessity to re-review.

Line 174-175: “Surprisingly, the SWI/SNF or BAF complex has not been shown to have a significant role in regulation of the rDNA locus”; briefly explain the connection between BAF and SWI/SNF here or in the preceding paragraph. Some readers may not know that BAF is a SWI/SNF type complex. BAF should also be listed in the abbreviation glossary.

Response: We thank the reviewer for their overall encouraging comments on the manuscript. With respect to the comment on Line 174-175, we expanded this introductory paragraph on the four classes of the chromatin remodeling complexes, since there are additional complexes which do not participate in rDNA regulation and add further context to the review. In addition, we have defined BAF and explained that it is the human equivalent to the yeast SWI/SNF complex.

In the cartoon figures (e.g., Figure 4), some of the protein names in the protein representations are hard to read, consider larger and bold font and maybe black instead of white. The histones (H3) in Figure 4 also look odd, like proteins sitting on top of DNA. A more ‘correct’ and appealing representation of nucleosomes with their histones might be good.

Response: We thank the reviewer for pointing out this oversight. As well as increasing the font size on the labeling of our figures, we divided figure 4 into separate figures (now figures 3-6) and placed them closer to the relevant text (as suggested by Reviewer 3). While redoing the figures we added a more accurate depiction of nucleosomes with DNA wrapped around them.

Reviewer 2 Report

Comments and Suggestions for Authors

This is a very well-written and comprehensive review work by the authors - Levesque and Picketts. The review nicely introduced the role of transcription factors and essential chromatin remodellers regulating rDNA structure and transcriptional status. Moreover, the authors reviewed the altered functions of these factors under relevant diseased conditions. Although it's not essential, however, it would be nice to add a section on the interconnection between these rDNA-specific chromatin remodelers and DNA damage repair on rDNA. Also, the authors may include a Table or a brief section addressing how evolutionary conserved these rDNA remodelers are across different species. Apart from those optional improvements, the manuscript can be accepted in its present form.

Author Response

Reviewer 2

This is a very well-written and comprehensive review work by the authors - Levesque and Picketts. The review nicely introduced the role of transcription factors and essential chromatin remodellers regulating rDNA structure and transcriptional status. Moreover, the authors reviewed the altered functions of these factors under relevant diseased conditions. Although it's not essential, however, it would be nice to add a section on the interconnection between these rDNA-specific chromatin remodelers and DNA damage repair on rDNA. Also, the authors may include a Table or a brief section addressing how evolutionary conserved these rDNA remodelers are across different species. Apart from those optional improvements, the manuscript can be accepted in its present form.

Response: We thank the reviewer for their positive feedback on our manuscript. We agree that a section addressing the connection between these chromatin remodelers and DNA damage repair would be informative, however we have chosen to focus this review on transcriptional regulation. Nonetheless, in Section 5.0 Conclusions and future perspectives, we have added the following sentence, “In this review, we have focused on the maintenance of rDNA transcription but it should be noted that many chromatin remodelers also participate in DNA damage repair on rDNA, which have been addressed in several excellent reviews.” These reviews have also been added to the reference list.

Reviewer 3 Report

Comments and Suggestions for Authors

GENERAL COMMENTS

The manuscript presents a comprehensive and very detailed overview about chromatin remodeling proteins that regulate ribosomal DNA expression in norm and disease. This is a needed review and is potentially of broad interest.

However, the manuscript is difficult to read and does not have a good flow. The authors need to organize the manuscript better and improve integration between different sections, as well as between the text and figures. In addition, references are missing from many places where they should be, including figure legends.

SPECIFIC COMMENTS

1.      The authors do not mention 45S rRNA at all. Why?

2.      Lines 59-61: sentence starting “Despite….” requires reference.

3.      Table 1 needs a heading. Also, the protein complexes included in Table 1 are described in the text much later. Thus, for the reader, it is difficult to follow what is the Table 1 all about.

4.      Lines 68-71: text about 5S rRNA requires reference.

5.      Please check that all acronyms are explained at first appearance. For example, TFs appear on line 118 but are not explained, likewise BAF on line 180.

6.      Section 2.3, starting line 139: ATP-dependent chromatin modelers are mentioned but not properly described nor provided with references, except ATRX. Though, all these protein complexes are described in detail in section 3. Please reorganize the text and connect to the data in Table 1.

7.      Please be consistent in section numbering: either 2, 3, 4 or 2.0, 3.0, 4.0.

8.      Page 7, lines 229-232: section starting “Werner and Bloom ……” is not connected with the rest of the text.

9.      All figure legends require references.

10.  All figures need to be properly, at all appropriate places, referred to in the text.

11.  Figure 1: acronym TSS not explained; black horizontal arrows not explained.

12.  Figure 2: RBS not explained; the legend explains only the lower part of the figure but not the upper part (comes in the next paragraph). Please revise the figure legend so that all elements and parts are properly explained.

13.  Figure 3: resolution of graphics not good; the legend needs to be more elaborated and the text needs revisions for syntax.

14.  Figure 4:  it is weird to have this figure in “Conclusions”, should be integrated with the main text; in figure legend “(A) NuRD (left) and CSB (right)” should be “(A) NuRD (top) and CSB (bottom)”.

15.  Page 12, line 509: NoRC complex is in Figure 4B and not Figure 4A.

16.  Page 12, line 510: reference to Figure 4B,C should be Figure 4A.

17.   Page 12, line 513: Figure 4D should be Figure 4C.

Comments on the Quality of English Language

Some revisions needed. 

Author Response

Reviewer 3

The manuscript presents a comprehensive and very detailed overview about chromatin remodeling proteins that regulate ribosomal DNA expression in norm and disease. This is a needed review and is potentially of broad interest.

However, the manuscript is difficult to read and does not have a good flow. The authors need to organize the manuscript better and improve integration between different sections, as well as between the text and figures. In addition, references are missing from many places where they should be, including figure legends.

SPECIFIC COMMENTS

  1. The authors do not mention 45S rRNA at all. Why?

Response: In humans, the fully unprocessed rRNA transcript is referred to as 47S rRNA. This is the first precursor that is transcribed, and this is what we refer to in the text. 47S rRNA is processed very early in the rRNA processing pathway into 45S rRNA by nucleolytic cleavage which partially removes the 5’ ETS and all of the 3’ETS sequence.

  1. Lines 59-61: sentence starting “Despite….” requires reference.

Response: We have added the necessary reference for this statement.

  1. Table 1 needs a heading. Also, the protein complexes included in Table 1 are described in the text much later. Thus, for the reader, it is difficult to follow what is the Table 1 all about.

Response: We agree with the reviewer that the Table was misplaced. We have added the necessary table heading and moved the table to the end of Section 2.3 where we first introduce all the remodeling complexes.

  1. Lines 68-71: text about 5S rRNA requires reference.

Response: We have added the necessary reference for this statement.

  1. Please check that all acronyms are explained at first appearance. For example, TFs appear on line 118 but are not explained, likewise BAF on line 180.

Response: We have reviewed the text and ensured the acronyms are all explained at first appearance and we updated our glossary of terms.

  1. Section 2.3, starting line 139: ATP-dependent chromatin modelers are mentioned but not properly described nor provided with references, except ATRX. Though, all these protein complexes are described in detail in section 3. Please reorganize the text and connect to the data in Table 1.

Response: We thank the reviewer for this comment which was similar to a comment by Reviewer 1. We have now expanded this section (2.3) to provide a better description of the four main classes of chromatin remodelers and the different complexes they form. We also moved the Table to the end of this section and provided a reference to the Table, so that it is easier for the reader to refer to it when looking over the different complexes.

  1. Please be consistent in section numbering: either 2, 3, 4 or 2.0, 3.0, 4.0.

Response: We have updated the section numbering to be consistent (2.0, 3.0, etc).

  1. Page 7, lines 229-232: section starting “Werner and Bloom ……” is not connected with the rest of the text.

Response: We agree with the reviewer and have removed this section from the text.

  1. All figure legends require references.

Response: We have reviewed our figure legends and added references to all of them where necessary.

  1. All figures need to be properly, at all appropriate places, referred to in the text.

Response: We thank the reviewer for this point. For clarity, we have broken up Figure 4 into four separate figures and moved them closer to the text where they are discussed. In addition, we reviewed the test and added reference to the figures where appropriate.

  1. Figure 1: acronym TSS not explained; black horizontal arrows not explained.

Response: We have updated the figure legend to indicate that the black horizontal arrows represent the transcriptional start site (TSS) of each repeat and at the same time defined the TSS acronym.

  1. Figure 2: RBS not explained; the legend explains only the lower part of the figure but not the upper part (comes in the next paragraph). Please revise the figure legend so that all elements and parts are properly explained.

Response: We thank the reviewer for pointing out this mistake. We have updated the figure legend to define the RFB acronym and we also added the acronym to the text at the start of Section 2.1 so that there should be no confusion. The figure has also been updated with all elements defined.

  1. Figure 3: resolution of graphics not good; the legend needs to be more elaborated and the text needs revisions for syntax.

Response: We have remade the figure and it is now much higher resolution. We have also rewritten the legend to better reflect the new figure.

  1. Figure 4:  it is weird to have this figure in “Conclusions”, should be integrated with the main text; in figure legend “(A) NuRD (left) and CSB (right)” should be “(A) NuRD (top) and CSB (bottom)”.
  2. Page 12, line 509: NoRC complex is in Figure 4B and not Figure 4A.
  3. Page 12, line 510: reference to Figure 4B,C should be Figure 4A.
  4. Page 12, line 513: Figure 4D should be Figure 4C.

Response: We thank the reviewer for this comment. Our original idea was that Figure 4 could serve as a summary figure of how the different complexes function at rDNA. We recognize that part of figure 4 was essential a duplicate of figure 3, so to better integrate the figure with the main text of the article we have split this figure into four new figures (one of which replaces figure 3). The separate figures now align much better with the text which we hope adds more clarity to the review.

Round 2

Reviewer 3 Report

Comments and Suggestions for Authors

The authors have adequately responded to the critique and revised the manuscript accordingly. No further comments.